# Flood Flow in a Proglacial Outwash Plain: Quantifying Spatial Extent and Frequency of Inundation from Time-Lapse Imagery

Clemens Hiller [1,2,*], Lukas Walter [1], Kay Helfricht [2], Klemens Weisleitner [3] and Stefan Achleitner [1]

1 Unit of Hydraulic Engineering, Department of Infrastructure, University of Innsbruck, 6020 Innsbruck, Austria; lukas.walter@student.uibk.ac.at (L.W.); stefan.achleitner@uibk.ac.at (S.A.)
2 Institute for Interdisciplinary Mountain Research, Austrian Academy of Sciences, 6020 Innsbruck, Austria; kay.helfricht@oeaw.ac.at
3 Institute of Ecology, University of Innsbruck, 6020 Innsbruck, Austria; klemens.weisleitner@uibk.ac.at
* Correspondence: clemens.hiller@uibk.ac.at

**Abstract:** High mountain environments have shown substantial geomorphological changes forced by rising temperatures in recent decades. As such, paraglacial transition zones in catchments with rapidly retreating glaciers and abundant sediments are key elements in high alpine river systems and promise to be revealing, yet challenging, areas of investigation for the quantification of current and future sediment transport. In this study, we explore the potential of semi-automatic image analysis to detect the extent of the inundation area and corresponding inundation frequency in a proglacial outwash plain (Jamtal valley, Austria) from terrestrial time-lapse imagery. We cumulated all available records of the inundated area from 2018–2020 and analyzed the spatial and temporal patterns of flood flows. The approach presented here allows semi-automated monitoring of fundamental hydrological/hydraulic processes in an environment of scarce data. Runoff events and their intensity were quantified and attributed to either pronounced ablation, heavy precipitation, or a combination of both. We detected an increasing degree of channel concentration within the observation period. The maximum inundation from one event alone took up 35% of the analyzed area. About 10% of the observed area presented inundation in 60–70% of the analyzed images. In contrast, 60–70% of the observed area was inundated in less than 10% of the analyzed period. Despite some limitations in terms of image classification, prevailing weather conditions and illumination, the derived inundation frequency maps provide novel insights into the evolution of the proglacial channel network.

**Keywords:** mountain hydrology; inundation; sediment transport; bedload; glacier; outwash plain; time-lapse; climate change

## 1. Introduction

Glaciers in high mountain areas are undergoing drastic alterations at unprecedented and increasing rates as an impact of global climate change [1–3]. The rapid mass loss of glaciers severely affects catchment hydrology and runoff dynamics in alpine mountain areas. The disintegration of glaciers and the decay at glacier tongues expose unconsolidated sediments in rapidly emerging proglacial areas. While these make up only a small areal proportion of alpine catchments, they are a significant source of sediments for mountain rivers [4]. The transition from glacial to deglacial conditions causes highly intensified geomorphic processes in the proglacial zones and freshly deglaciated areas [5,6]. Typically, most sediment export from glaciated catchment in the Alps occurs in the months of July and August [7]. At the same time, the overall glacier area is decreasing, where potential ice melt contributes to runoff. Thus, peak runoff in summer caused by glacier melt has been observed to decline or is assumed to do so over the next decades. In the Alps, a significant part of today's glaciers may disappear within this century [8,9], leading to a fundamental shift in runoff regimes [10]. Particularly, a general decrease in runoff during summer months is predicted [11]. At the same time, an increase in extreme (convective) rainfall

events is expected [12,13], partly driven by increased evapotranspiration from the surface. In this context, the anticipated progressive system shift from supply-limited sediment transport (driven by glacier activity) to more transport-limited processes (during rainfall-induced events) in high alpine catchment areas has been the subject of recent scientific debate [14]. Especially in glacier forefields, sediments are deposited in the outwash plains under average runoff conditions, but during high/extreme discharge, re-mobilization can occur, resulting in high rates of sediments transported downstream. Therefore, outwash plains are key elements in high alpine river systems and promise to be revealing, yet challenging, areas of investigation for the quantification of proglacial fluvial processes. Contemporary deglaciation offers an unparalleled opportunity to directly observe current processes and mechanisms of sediment transport and basin-scale sediment storage and release [15]. In published examples [16], up to 90% of the proglacial area is estimated to be affected by glacial runoff. However, the extent of the inundated area in proglacial outwash plains and the frequency of inundation is largely unknown because of the typically remote location of outwash plains in the proximity of retreating glaciers. These environmental conditions usually complicate more direct measurements, especially since the object of investigation is prone to frequent geomorphic processes and changes.

In this article, we demonstrate the applicability of a methodological approach to constrain spatial and temporal dynamics of surface runoff in a sample outwash plain (Jamtal valley, Austria) to overcome data scarcity on paraglacial processes. The approach utilizes a terrestrial time-lapse camera with a high-resolution RGB-sensor. Previous research has proven the value of oblique digital imagery to monitor the planform, topography and rates of change in braided river channels [17–19], as well as discharge patterns in remote, steep and boulder-strewn channels [20,21]. Remote sensing techniques cover large-scale, relatively long duration flood events, e.g., [22], while relatively small, confined (urban) areas are characterized by high sensor density and data availability, e.g., [23,24]. However, remote and therefore often data-poor environments at the sub-catchment level remain insufficiently surveyed due to the lack of available sensors and adequate resolutions, e.g., [25,26]. Terrestrial time-lapse imaging platforms are therefore a suitable means to close this monitoring gap.

The aim of this study has been to capture the runoff and floodplain dynamics over a three-year observation period in this alpine proglacial outwash plain based on terrestrial time-lapse imagery. We have focused on the following questions:

- To what extent can semi-automated image analysis capture the inundation of an active floodplain?
- Which processes generate the most extensive observed inundated area?
- How strongly does the channel network alter inter-annually and during high-runoff events?

Furthermore, we present the derived and geo-rectified inundation frequency maps, depicting the annual cumulation of all available and classified images, plus a detailed analysis of the daily maximum inundation area ($IA_{dmax}$). In this way, we offer a means for a better understanding of flood flow and runoff behaviour of the here-investigated type of alpine proglacial outwash plains.

## 2. Materials and Methods

Our research was conducted in the proglacial outwash plain of Jamtalferner glacier (Video S1) in southwestern Tyrol, Austria. The Jamtal valley, and especially the rapidly retreating Jamtalferner glacier [27], have long been subject to environmental investigations and are part of the Austrian Long-Term Ecosystem Research Network (LTER-Austria) [28]. Several measuring stations that record various key parameters on a catchment scale (Figure 1a–c) are located on site. A time-lapse camera in the immediate glacier foreland as well as an automatic weather station (AWS) and a radar-based discharge gauge a few kilometers down the valley were of central importance for the work presented here.

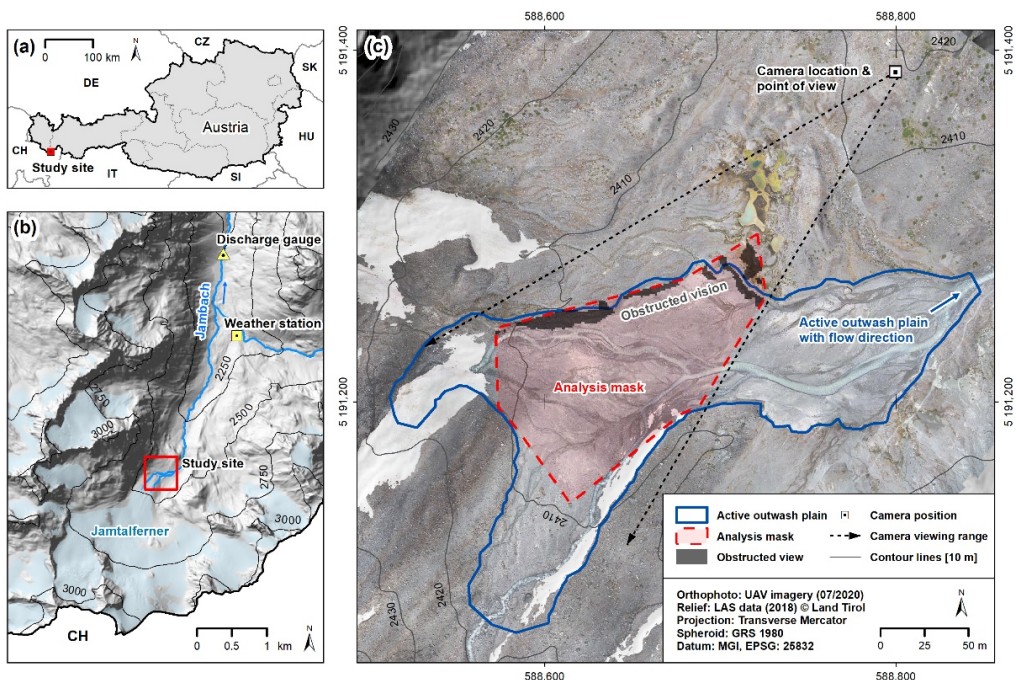

**Figure 1.** Overview of the study site at the head of the Jamtal valley. The investigated area is (**a**) located in the west of the Austrian state of Tyrol and borders on Switzerland. It is (**b**) equipped with an automatic weather station (2141 m a.s.l.) and a radar-based discharge gauge (~2010 m a.s.l.) at the Jambach stream. (**c**) shows an active outwash plain (solid dark blue line, ca. 2.8 ha at ~2410 m a.s.l.) near the progressively disintegrating Jamtalferner glacier. The southwest-facing time-lapse camera (indicated with black dotted lines) overlooks the upper half of the investigated outwash plain with its two main tributaries that emerge from the partially debris-covered glacier tongue. The section analyzed in the time-lapse imagery is highlighted by the dashed red line. Areas of obstructed vision induced by topographical obstacles are indicated in dark grey.

### 2.1. Time-Lapse Camera

The time-lapse images were captured with a Nikon D300 digital single-lens reflex camera mounted inside a waterproof housing set up on top of a boulder near the glacier forefield (Figures 1c and 2). While the camera aperture (f 8.0), white balance (5560 K), sensor gain (ISO 200) and focus (infinite) were set manually, changing light conditions required a variable integration time that was automatically adjusted by the camera.

Both the power supply and the shutter release were controlled by an Arduino Uno microcontroller connected with a real-time clock (model: DS3231) and powered by a 12 V 17 Ah battery that was charged with a 100 W solar panel. Based on a simple loop function, the camera was switched on by activating a power relay module connected to the 12 V battery and a voltage regulator that provided a stable 9 V power supply for the camera. Then, a 5 V signal was sent to the external shutter port of the camera and released the trigger. The image was saved locally on a SD memory card (32 GB) connected to a CF card adapter before the relay interrupted the camera power supply. This loop (i.e., switching on the camera, capturing and saving an image, switching off the camera) was active daily at an hourly interval during observation periods. Moreover, camera recordings started at 6 am CET and ended at 8 pm CET to reduce the number of night images.

### 2.2. AWS—Automatic Weather Station

The meteorological data were measured with an AWS located in the vicinity of the Jamtalhütte mountain hut at 2141 m, about 2 km from the investigated outwash plain. The AWS had been operated since 2013, and the data were provided by the Hydrographic Service, Regional Government of Tyrol. We calculated hourly values and the daily maxima for the measured variables of air temperature, global radiation and precipitation to align

the meteorological data for analyses and comparison with the hourly time-lapse results from the outwash plain.

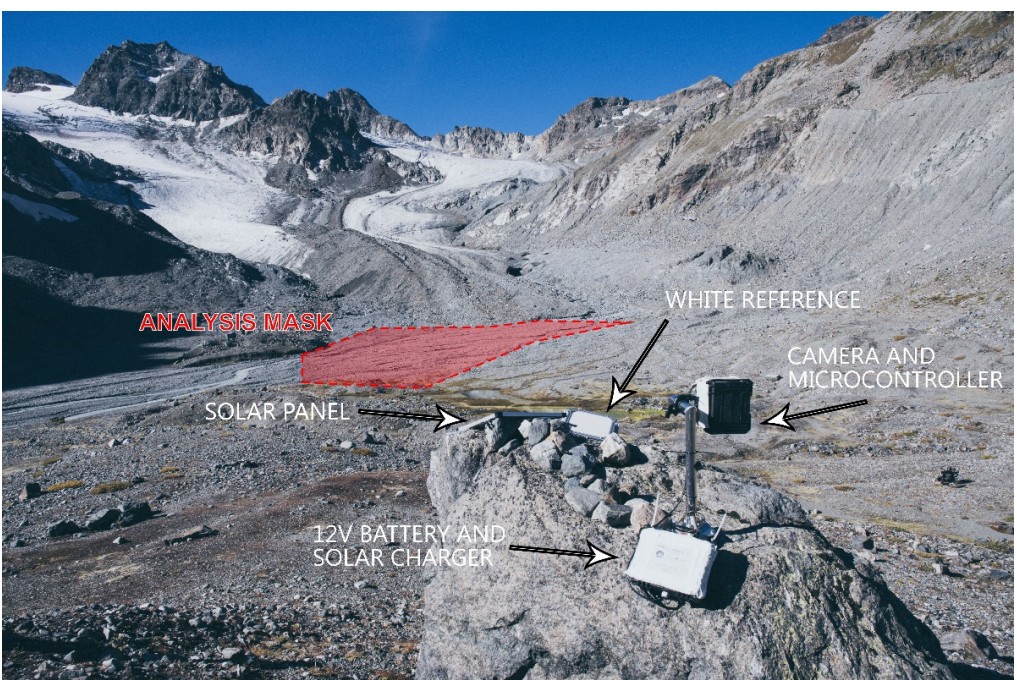

**Figure 2.** Setup of the time-lapse camera near the Jamtalferner glacier (view to south-west), mounted on top of a large boulder overlooking the investigated proglacial outwash plain highlighted in red.

### 2.3. Discharge Gauging Station

Discharge estimates based on water level and surface velocity measurements were established at a bridge across the Jambach stream in June 2019. This gauging station was located about 3 km from the investigated outwash plain, at an elevation of ~2010 m a.s.l. and provided continuous data for the avalanche-free season, which usually lasts from June to November. In this time frame, the station covered the runoff pattern from the onset of the high mountain snowmelt in spring until the discharge stagnated again at a low level in autumn. The runoff gauge was equipped with an RQ-30 device from Sommer Messtechnik (Koblach, Austria) [29]. The RQ-30 is a non-contact flow velocity and water level sensor with an integrated data logger and data transmission bundled in a compact housing. During the three summer seasons from 2019 to 2021, 20 salt dilution measurements were carried out and used to calibrate stage-velocity/discharge relations.

### 2.4. Image Analysis Process

The workflow applied for the image analysis is shown in Figure 3 as a schematic flowchart. In a first step, the annual subsets of digital RGB color images were manually examined for defects and quality to meet storage space requirements for the ensuing auto-mated analysis process. This was followed by an automated quality filtering in the image processing program Fiji, a distribution of the open-source software ImageJ (version 2.0.0-rc-69/1.53c; Java 1.8.0-172) [30]. Therein, user-defined macros enabled image processing and analysis. Consequently, we designed back-to-back macros (Scripts S1) capable of extracting pixel-based information to identify inundated areas in the camera view.

In a first step, images were filtered by means of the given values for RGB- and greyscale-channels using the *SORT OUT* algorithm in ImageJ. As an initial constraint (P1), only images with greyscale values (GV) between 14 and 100 were retained and saved for subsequent processing. These thresholds for the median of the GV were determined based on single image analyses, which identified images to be excluded that were taken either in the presence of a closed snow cover (GV > 100) or in insufficient light (GV < 14). A second

constraining parameter (P2) was set with a minimum value of −1.00 for the kurtosis in the central area of the RGB images covering the outwash plain to detect and dismiss images with partial snow cover or taken under foggy conditions. A third parameter (P3) was used to reject images taken under foggy conditions or with a wet lens, assessing the standard deviation (SD) of the GV and setting the lower limit to 15. Images falling below the threshold value were excluded.

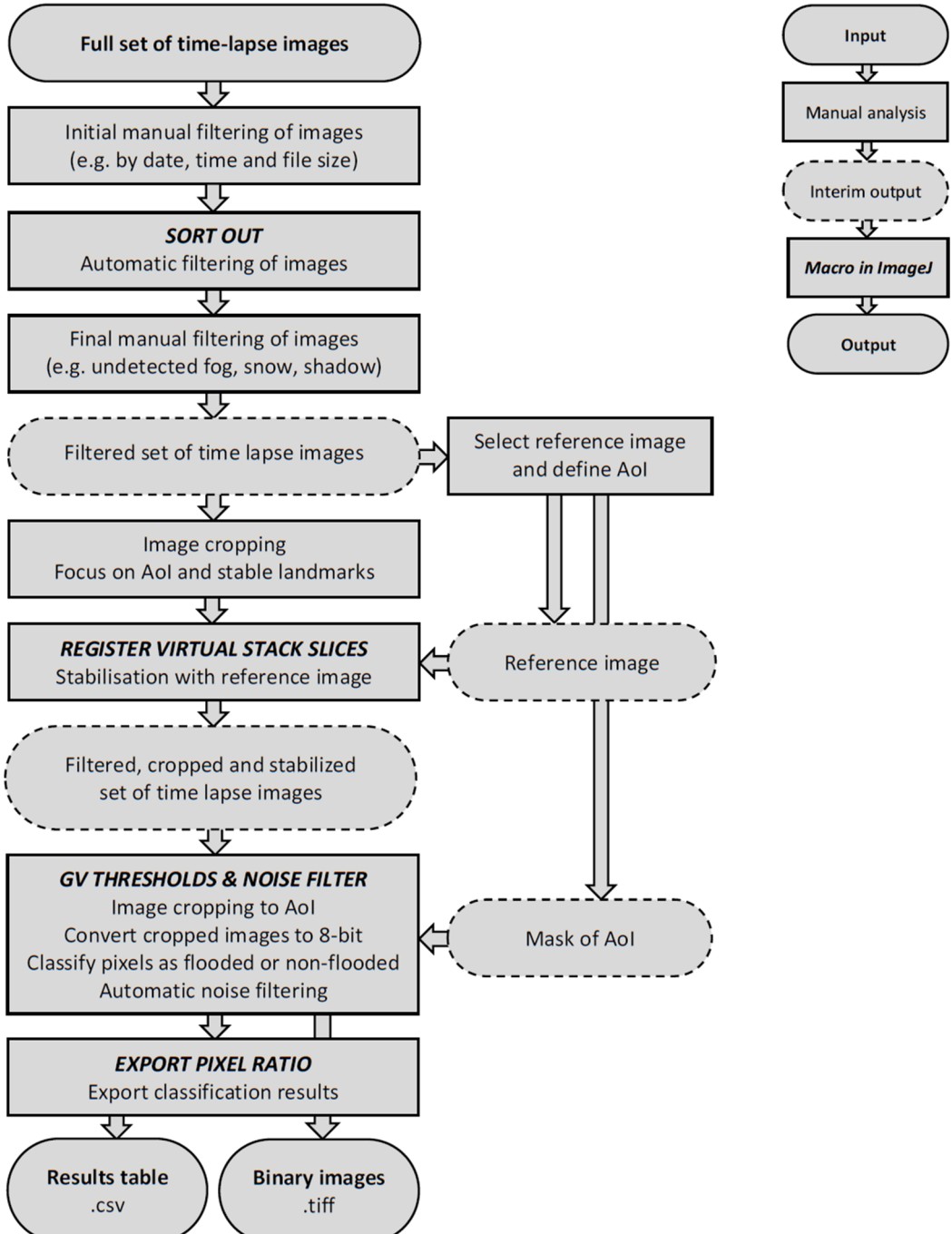

**Figure 3.** Schematic workflow of the semi-automatic process of image analyses. Boxes with capitalized labels refer to customized macros used in ImageJ.

Comparing the medians of the GV in the right and left half, as well as the upper and lower half of the images, provided a fourth parameter (P4) to exclude images with uneven lighting and extensive shadow in the outwash plain. The thresholds for a maximal median

offset for the comparison of the right and left image halves were set to Δ15 and for the median comparison of the upper and lower halves to Δ20. All images with a median offset exceeding these figures were dismissed under the assumption of unsuitable lighting conditions.

Secondly, all remaining images were being cropped by a mask that excised the lowest quarter depicting a white balance plate mounted on the boulder and fluvially inactive areas at the margin of the outwash plain. This eliminated some of the potential sources of disturbance during the ensuing image stack stabilization and further reduced the image set's total data size. In the case study presented, the initial imagery consisted of over 2000 images per year, while the remaining sets were reduced to n = 1168 images in 2018, n = 180 in 2019 and n = 900 in 2020 (Table 1). The remaining images defined the beginning and end of the three observation periods. The set of 2019 yielded the fewest images due to a malfunction of the camera over several weeks in the summer season.

**Table 1.** Observation periods and annually available images before (total) and after (sorted) applying the semi-automatically sorting algorithm. The reference image corresponding to the respective observation period that was used for stabilization is shown with the date and time.

| Year | | 2018 | 2019 | 2020 |
|---|---|---|---|---|
| Total | Records | 2844 | 2123 | 2939 |
| | Time period | 26 May–16 December | 1 February–31 December | 1 January–25 October |
| Sorted | Records | 1168 | 180 | 900 |
| | Time period | 12 July–16 October | 18 September–1 November | 8 July–24 September |
| Reference image | | 31 July 9:50:10 | 1 October 6:55:32 | 21 August 12:52:20 |

Image stabilization is an essential step to ensure the comparability of a pixel-based analysis. The idea is to link any static object or structure to a specific and static coordinate system within the stabilized image set. In the presented work, the stabilization was achieved with the *ImageJ* plug-in REGISTER VIRTUAL STACK SLICES [31]. The set of images was regarded as a virtual stack, with the images of the stack called slices. The plug-in was used to align the virtually stacked image slices by means of translation and rotation to a reference image from the same stack, resulting in a constant image size.

The stabilization for the presented imagery was carried out with annually separated image sequences (2018, 2019, 2020), since the image stabilization did not produce a sufficiently stable set of images for all three observation periods combined. The shift in the image orientation between different years proved to be too substantial to be corrected and stabilized by translation and rotation alone.

Still, identical stabilization settings were used, processing the three, annually separated virtual stacks. The relevant reference image was selected manually from each of the corresponding set of images (Table 1), considering the best possible lighting, visibility and snow-free conditions for efficient image feature matching.

The feature extraction model that derived corresponding points from the images was set as rigid, the same as the registration model, limiting the types of transformation to translation and rotation. Otherwise, all the default settings of the plug-in were used for the image stack stabilization.

The subsequent step utilized the macro GV THRESHOLDS & NOISE FILTER and was split into four steps. Using the already sorted and stabilized images, (A) image cropping was followed by (B) image conversion to 8-bit, (C) pixel classification and (D) automatic noise filtering. Step (A) limited the investigated area, since long-lasting snow fields were still visible in parts of the images, predominantly occupying the image's borders. To prevent snow fields from disturbing the detection of flooded areas, a mask was defined, limiting the detection area. It was set to cover an assumed maximum extent of the potential inundation area (IA) and to exclude areas with long-lasting snowfields. To create a consistent mask for the outwash plain, landmarks in the terrain, such as large boulders, were chosen as corner points. After the conversion of each image to 8-bit greyscale (B), a classification scheme to distinguish between flooded and non-flooded areas was applied (C). Therefore,

approximately 50 representative images from the available recordings were analyzed. The flooded areas in these images were separated by means of visual inspection and built the basis to develop a classification scheme. Since all the selected pictures featured varying lighting conditions, setting simple (fixed) upper/lower limits ($y_{GV,U}/y_{GV,L}$) to identify pixels as flooded/non-flooded areas proved to be unsuitable. Considering the overall variation of GV in each picture, the median of the greyscale values ($x_{GV}$) was found to be sufficient to define the specific (image-wise) settings of the upper and lower threshold (1).

The dependency is described by a third-order polynomial function:

$$y_{GV} = a_3 \cdot x_{GV}^3 + a_2 \cdot x_{GV}^2 + a_1 \cdot x_{GV} + a_0, \tag{1}$$

with individual coefficients set to describe the upper and lower thresholds ($y_{GV,U}/y_{GV,L}$) of GV:

$$y_{GV,U} = 0.00064 \cdot x_{GV}^3 - 0.074 \cdot x_{GV}^2 + 3.73 \cdot x_{GV} + 96.2 \tag{2}$$

$$y_{GV,L} = -0.00022 \cdot x_{GV}^3 + 0.027 \cdot x_{GV}^2 + 0.18 \cdot x_{GV} + 33.6. \tag{3}$$

The two Functions (2) and (3) automatically adjusted for each image used, and therefore, accounted inherently for varying lighting conditions. Consequently, no white-balance or any other adjustment of the image exposure was required. All pixels with a GV within the thresholds ($y_{GV,U} > y_{GV,i} > y_{GV,L}$) were considered flooded areas and were converted into white. All remaining, non-flooded parts were set to black.

In the last step (D), noise filtering was applied. The procedure for outlier removal was applied to pixels identified as a flooded area. Still, the algorithm was based on the 8-bit greyscale image, where a pixel was compared to its vicinity. Outliers were removed in case the GV of the respective pixel deviated from the median value of the neighboring pixels more than a given threshold. Here, we applied a filtering radius of three pixels and a maximum deviation from the median GV of $D_{GV} = 50$. This way, other reflecting objects, e.g., bright or wet cobbles or small patches of thin snow, which might have been misidentified as water surface, could be corrected. The final images, classified and noise-filtered, were saved in TIFF-format (8-bit).

The areal inundation ratios were calculated for further time series and statistical analysis with the macro *EXPORT PIXEL RATIO* and exported as .csv-files (Table S1).

## 3. Results

### 3.1. Images Available for Analysis

In Figure 4, it is apparent that the hourly recordings of the camera, which were evaluated after pre-processing, were distributed almost evenly throughout the day. Due to the camera settings, the recordings were available from 6 am to 8 pm CET and were thus limited to the daytime with good visibility. The highest number of images was obtained for 2 pm CET in 2018 and for 3 pm CET in 2020. The higher number of recordings at 6 am CET compared to the following hours was likely caused by favorable lighting conditions without direct insolation and thus without shadows in the images. In contrast, the number of records decreased in the final hours to 8 pm CET, which was due to the shift of dusk to earlier times in late summer and autumn. A notable feature of the 2019 dataset was the lower share of total images used in final analyses, which was due to adverse weather conditions and intermittent camera malfunctions.

### 3.2. Areal Inundation Frequency

The analyzed time-lapse images were cumulated to a single inundation map for each observed year. The cumulative number of wetted surface observations per pixel and year can be interpreted as a flooding frequency.

Finally, these maps were rectified to the topographical model of the outwash plain, visualizing the extended and spatially distributed frequency of surface flows captured per observed year (Figure 5). Inter-annually, the extent of the analysis mask varied slightly, as the lateral snowfields were cropped out to prevent misclassifications by the image analysis process

(e.g., snow surface as a flooded area). However, the strikingly different appearance of Figure 5b, in contrast to Figure 5a,c, can be explained by the differences in the season considered as well as the severely limited image availability for the year 2019 (also see Table 1).

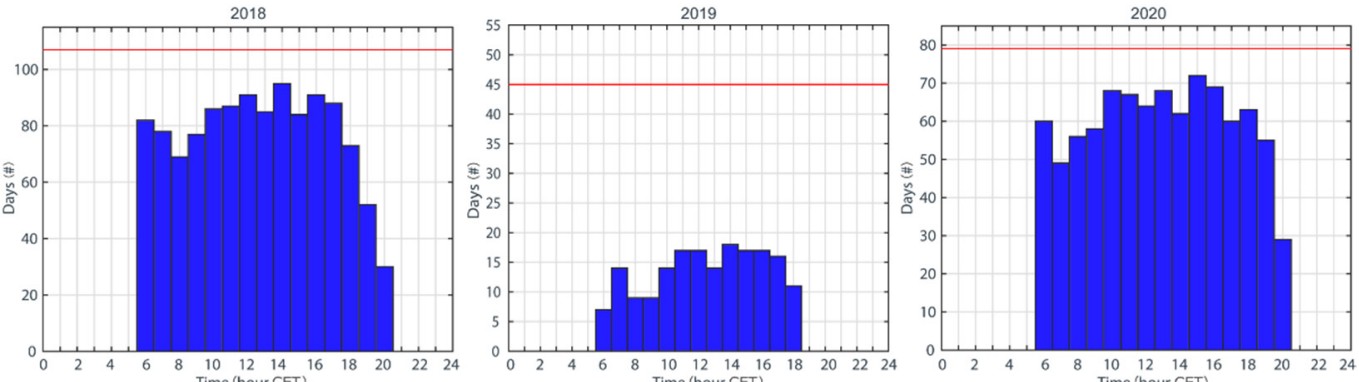

**Figure 4.** The number of days with recordings for a given hour of the day in the final analysis after sorting and processing. The red line shows the maximum number of days between the first and last recordings of the respective year.

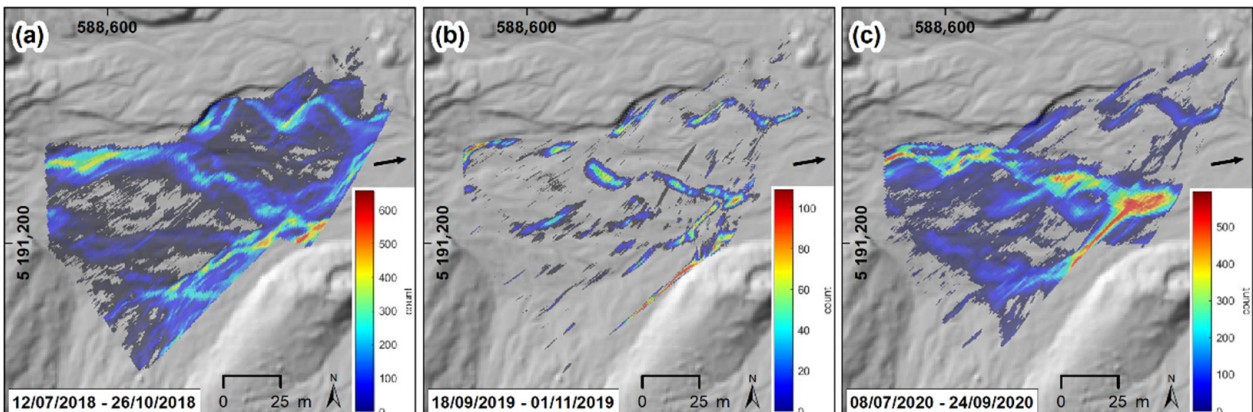

**Figure 5.** Inundation frequency maps for the three observation periods of (**a**) 2018, (**b**) 2019 and (**c**) 2020 in the proglacial outwash plain of the Jamtalferner. The maps show the total number of times (analyzed images) that a location (pixel) was identified as a flooded area. The general direction of flow is indicated with a black arrow. Note: the maximum values differed among the maps due to deviating observation periods and to the final availability of the images after the sorting process.

For the period in 2018 (Figure 5a), a well-defined channel network was visible, with two main streams emerging from the glacier tongue. The northern stream furcated into two subsections with approximately the same inundation frequency. The most frequent flooding occurred at the convolution of the main streams on the eastern boundary of the analysis mask. Several less frequently flooded tributaries could also be seen and indicated the maximum extent of the channel network. During the period in 2019 (Figure 5b), channel patterns largely similar to the observation period of 2018 were observed, while less-frequently flooded tributaries remained undetected within the six observed weeks in autumn. The third observational period in 2020 (Figure 5c) revealed a shift in the inundation frequency pattern of the outwash plain. The main streams appeared to have remained fairly stable over the three analysis periods. However, the most northern subsection showed a relatively lower flooding frequency, while the central subsection had a higher flooding frequency than in 2018 and 2019.

In this context, the changes in areal inundation frequency between 2018 and 2020 could be described as an increased degree of channel concentration, i.e., the detected frequency

of flooding applied to a reduced IA as a result of the shift described above. This shift in the ratio between the cumulative share of the total area of interest (AoI) and the counts (pixels detected as covered by water) relative to the maximum counts became visible when considering that 5% of the IA accommodated about 35% of the counts in 2018, while by 2020, the comparative value had increased to approx. 50% (see also Figure A1). The observed concentration of surface runoff patterns reflected the evolution of the channel network as it altered the spatial and temporal properties of the IA between 2018 and 2020.

### 3.3. Daily Maximum Inundation Area

In addition to the spatial mapping of inundation frequencies on an annual basis, the daily maxima of the detected IA were compared with meteorological and discharge measurements (Figure 6, Table S2). While data on maximum air temperature and global radiation were provided for all three periods, runoff estimates were only available from summer 2019.

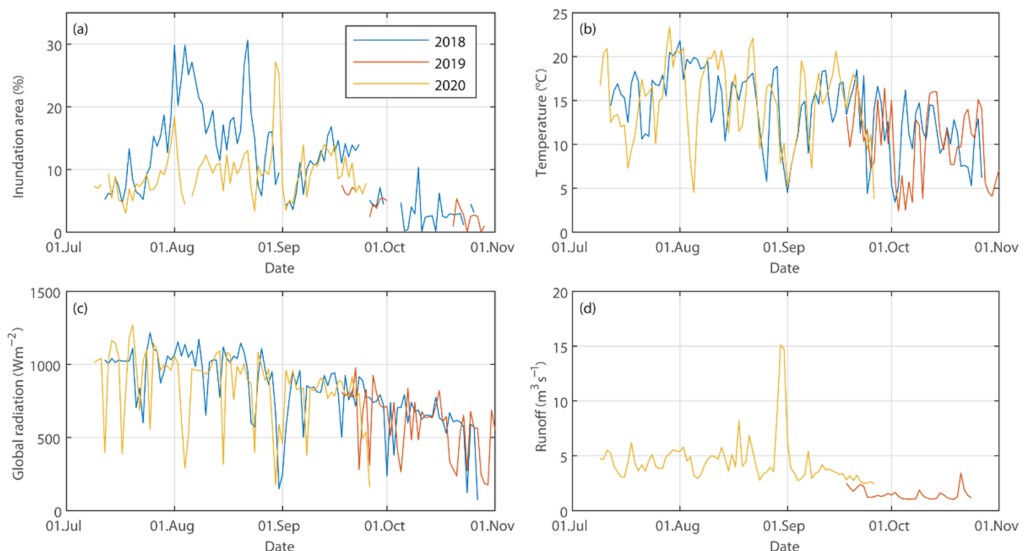

**Figure 6.** Daily maximum of (**a**) the detected inundation area relative to the AoI, (**b**) air temperature, (**c**) global radiation and (**d**) runoff for the three observation periods between 2018 and 2020 (also see Table S2).

In 2018, high percentage values for the daily maximum inundation area of up to 30% indicated extensive flooding of the outwash plain, caused by snow melt and ice ablation due to clear sky conditions and high temperatures in combination with surface runoff from precipitation in the afternoon (red highlighted in Figures 7 and 8) in early August.

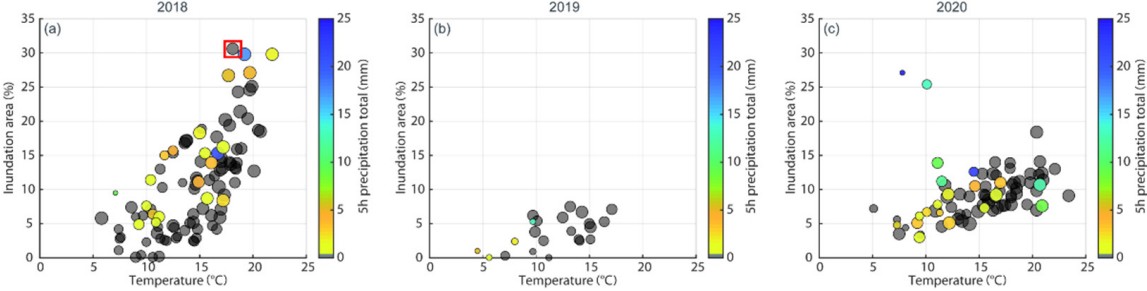

**Figure 7.** $IA_{dmax}$ plotted with day/month in the observation period. The colored scheme of the plot relates to the total precipitation occurring within 5 h prior to recordings in 2018 (**a**), 2019 (**b**) and 2020 (**c**). Additionally, the circles are sized according to the daily maximum global radiation. Note that the red highlighted circle in the 2018 data is discussed in the text and Appendix A.

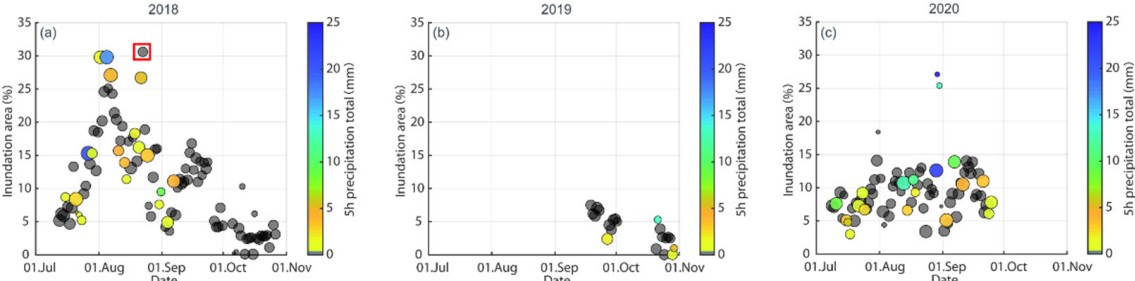

**Figure 8.** $IA_{dmax}$ plotted with the daily maximum air temperature; the colored scheme of the plot relates to the total precipitation occurring within 5 h prior to recordings in 2018 (**a**), 2019 (**b**) and 2020 (**c**). The circles are sized according to the daily maximum global radiation. Note that the red highlighted circle in the 2018 data is discussed in the text and Appendix A.

In contrast, a cold front at the same time in 2020 led to a drop in air temperature and global radiation, which significantly decreased ablation. The highest $IA_{dmax}$ in 2020 was recorded in late August with 27% as a result of a single, prominent flooding event due to heavy precipitation over three successive days caused by cold air advection to the Alps. These observations are supported by the gauged discharge data further downstream. The subsequent low values of $IA_{dmax}$ and runoff could be attributed to the low air temperatures and, thus, modest ablation in the wake of the event.

The comparatively shorter data series of 2019 covered the period of autumn, with characteristic minor runoff due to reduced global radiation and therefore less ablation on the glacier. The $IA_{dmax}$ values of 2019 exhibited a similar order of magnitude as the figures for the same time in 2020.

The inter-annual evolution of the inundation area is presented in Figure 7. In the 2018 data set, the highest $IA_{dmax}$ values of about 30% were recorded at the beginning of August. This coincided with high levels of meltwater discharge and additional convective precipitation, represented in the shape of large circles (high global radiation) and the precipitation-indicating coloring. Towards the end of August and throughout September, the $IA_{dmax}$ values fell consistently below 20%, while even days with high global radiation and convective precipitation did not result in increased $IA_{dmax}$ values. In addition, no precipitation was recorded from mid-September to the end of the observation period at the end of October, and global radiation generally decreased. As a logical consequence, the $IA_{dmax}$ values continued to decrease as the season progressed, to below 5% at the end of October.

This trend of decreasing $IA_{dmax}$ in the course of autumn was also visible in the 2019 data set. The $IA_{dmax}$ values were in the range of 5–10% in mid-September, while by the end of October, they had fallen to a maximum of 5%. It is noticeable that, in contrast to 2018, precipitation causing $IA_{dmax}$ was recorded late in the observation period. However, these precipitation events did not cause a significant increase in the detected spatial extent of the inundation area, as part of the precipitation was likely to have occurred as snow fall in the upper catchment.

The observational period in 2020 covered the months July to September and presented a rather different seasonal evolution. Generally, there was a slight increase in $IA_{dmax}$ over time, starting with values of mostly 5–10% in July, around 10% in August and finally up to 15% in September. Compared to the data series of 2018, in particular, the lower $IA_{dmax}$ values in the first half of the data series, with no significant peak in the spatial extent of the inundation area, indicate deviating snow conditions and melt for early summer 2020. Although most data points were between 5% and 15% throughout the 2020 period, there were outliers, particularly in July and August, below 5% and above 25%, respectively. It is notable that the outliers above 25% could be associated with low global radiation and high precipitation. A trend towards decreasing $IA_{dmax}$ values in autumn, as seen in 2018 and 2019, could not be derived for the 2020 data, since the respective observational period ended with September.

The observed spatial extent of the inundation area and its dependency on meteorological factors is further explored in Figure 8. The $IA_{dmax}$ value rarely exceeded 15% under clear sky conditions, with high ablation rates on the glacier. The higher values depicted in the 2018 data could be primarily related to excessive snow and ice melt in July. In general, $IA_{dmax}$ on days with measured precipitation coincided with lower maximum air temperatures, compared to $IA_{dmax}$ on typical ablation days. Moreover, most of the $IA_{dmax}$ values exceeding 25% in 2018 occurred on days with heavy ablation, triggered by high global radiation in the first half of the day, paired with additional precipitation in the afternoon or evening, resulting in distinctive flooding events in the investigated outwash plain.

The respective data of 2019 showed a comparably lower $IA_{dmax}$ that remained below 10%, representing a shorter observational period in late summer and autumn, with reduced global radiation and precipitation that started to fall as snow at higher altitudes of the upper catchment. The generally lower air temperatures in the observational period of 2019 also reflected the later season of the recordings.

In contrast to the two previous years, the two highest $IA_{dmax}$ values of over 25% in the data set for 2020 occurred at relatively low air temperatures and equally low global radiation but were triggered by heavy precipitation events.

For the 2020 data set, we also compared the 2020 $IA_{dmax}$ percentage with the daily maximum runoff measured at the gauge with respect to the time of day (Figure 9). A clustering around 4 pm could be seen on days without recorded precipitation prior to $IA_{dmax}$. The wider range and offset towards the measured daily maximum runoff between 5 pm and 8 pm may implicate a systematic underestimation of $IA_{dmax}$ in the late afternoon hours due to unfavorable lighting conditions. As could be expected, the time of day for the runoff maxima was delayed compared to the detection time of $IA_{dmax}$ because of the distance of ca. 3.5 km between the glacier forefield and the runoff gauge downstream. This may also explain the minor but continuous underestimation of $IA_{dmax}$ on days with significant precipitation events.

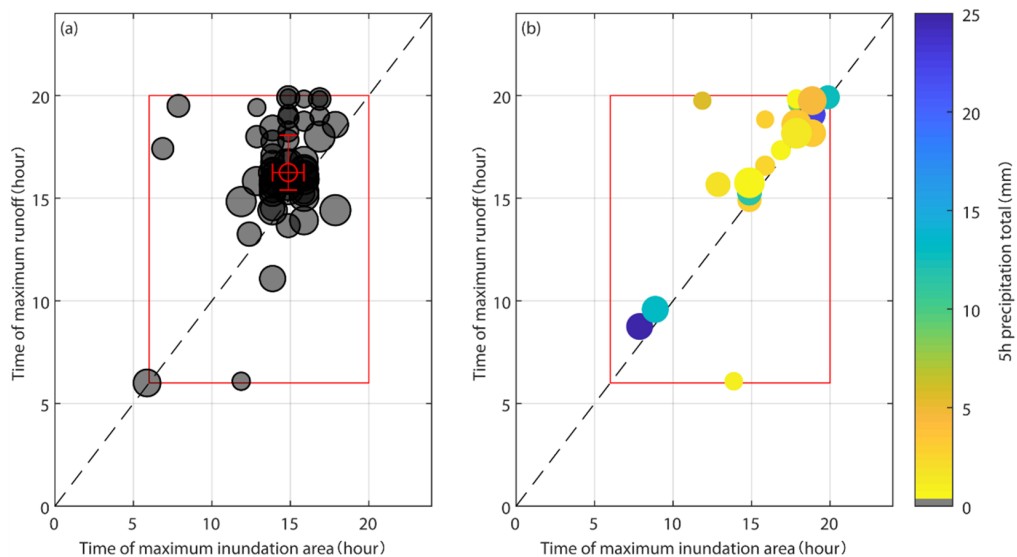

**Figure 9.** Time of the daily maximum inundation area ($IA_{dmax}$) plotted with the time of maximum runoff in 2020 for days (**a**) without precipitation and (**b**) with precipitation within 5 h prior to the recorded $IA_{dmax}$. The color scheme in (**b**) relates to the total precipitation 5 h prior to recordings. The circles are sized accordingly to the daily maximum values of global radiation. The red box marks the time frame of available camera recordings.

## 4. Discussion

The approach presented here of using a time-lapse camera provides insights into proglacial runoff dynamics that could otherwise only be quantified with considerable difficulty, e.g., [10]. The alternative of implementing a conventional gauging station was

dismissed due to the inherent dynamics of the braided river system with intense lateral channel shifting and geomorphic processes (e.g., lateral erosion, avalanches, rockfall) at potential station sites. Furthermore, widely used remote sensing techniques for flood detection have their limitations when applied in mountainous sub-catchments. Optical remote sensing data is strongly affected by cloud obscuration, restricting the ability to detect inundation induced by rainfall [25]. In contrast, the increasingly established application of a synthetic aperture radar (SAR) circumvents the issue of cloud cover. Nevertheless, the spatial resolution of 5–10 m and multi-monthly revisit-intervals of current non-commercial SAR satellites suggest that delineation of short-lasting floods in headwater catchments is not yet available with adequate robustness [26]. The utilization of terrestrial time-lapse imagery partly mitigates these limitations for inundation mapping in a 2.8 ha outwash plain. In this context, our estimations of the inundated area from image analysis were broadly convincing, especially for weather conditions with liquid precipitation. However, the here presented method entailed its own limitations that need to be addressed when discussing the advantages and weaknesses. In daylight, the approach was not limited to ideal conditions but provided reliable results for a wide range of light and weather conditions. However, the method's weak point was undoubtedly (in)sufficient lighting and whether patches of snow or fog obstruct the camera's view.

### 4.1. Daily Maximum Inundation Area

To obtain visual information on the extent of the inundation from an RGB camera, a sufficiently and uniformly illuminated (see Chapter 2, P4) observation area free of snow cover is required. For this reason, only images at daylight and without snow cover in the AoI could be analyzed. This entails the inherent problem of the method of being unable to capture high runoff events at night, for example, in the case of high meltwater runoff in conjunction with a thunderstorm in the late evening. In this context, precipitation-bearing convective clouds in the afternoon (Section 2, P1) could limit visibility to the extent that the affected images have to be excluded from further analysis. Furthermore, the strict exclusion of images with snow cover in the AoI (Section 2, P2) deprives the method of quantifying typically high runoff conditions due to snowmelt in the glacier forefield. Conversely, the AoI was defined to exclude known areas of persistent snow accumulation. However, it captured high runoff caused by continuous snow and subsequent ice melt at higher elevations of the glacier.

The inundation frequency maps (Section 3.2, Figure 5a–c), as annual cumulations of the observed daily inundation extent, could be interpreted as the evolution of the proglacial channel network over three years. A similar image availability for Figure 5a,c (representing 2018 and 2020) allowed a direct comparison of the flooding frequency of individual tributaries. In contrast, Figure 5b was based on significantly fewer images in a shorter observation period. Nevertheless, the inundation frequency map for the period in 2019 indicated a widely unchanged channel network compared to the observation period of 2018. Less-frequently flooded tributaries remained undetected within the six observed weeks, which could be attributed to typically decreased meltwater discharge and less frequent convective precipitation in autumn. The evident shift in the inundation frequency pattern within the outwash plain between 2018 and 2020 was presumably initiated with the onset of snow melt in early summer 2020 and subsequently continued by heavy precipitation events in the following months. However, the associated increased degree of channel concentration could not be attributed to a specific flood event due to the remaining complexity of the braided channel network.

### 4.2. Weather Conditions and Illumination

As discussed above, the presented method is highly dependent on the illumination of the pre-selected images. The degree and distribution of the scene luminance, in turn, was influenced by the weather conditions. In contrast to optical satellite data, in most cases, cloud cover was only an issue if it occurred as fog at the elevation of the camera location and

the outwash plain. Overall, our estimations of the inundated area from image analysis were broadly convincing, although some weather conditions were found to be more favorable to the method than others. A clear sky, for instance, caused a distinct shadow in early evening recordings and led to a systematic underestimation of the IA and thus to earlier $IA_{dmax}$ times. Moreover, uniformly underexposed images could also result in an underestimation of the IA, simply by determining an erroneous threshold (Figure A2). In contrast, a discrete, intensely illuminated area in an AoI that was otherwise covered by shadows was detected as inundated, as the example in Figure A3 demonstrates. In both cases, a further refinement of the filter constraints in the respective ImageJ macros could be beneficial to sort out such cases more efficiently.

Besides weather-dependent illumination variants, adverse effects from precipitation could account for data gaps and image misanalysis. If the solar panel of the time-lapse camera was covered by snow for more than two days, it resulted in a temporary battery drain. This circumstance created occasional data gaps that exceeded the number of days with snow cover due to the slight delay in recharging the camera and resuming the hourly recordings. We also observed that raindrops on the window of the camera housing affected the final results (Figure A4). The affected parts of the image were darker and blurred; thus, they were not considered in the IA count. This could lead to an underestimation of the IA during (heavy) precipitation events.

### 4.3. Spatial Segregation of AoI and AWS

We compared the IA results with meteorological data from the closest available AWS. However, the spatial distance between the AoI and the AWS (see Figure 1 in Section 2) caused some inconsistency in the results and the final interpretation. In particular, the maximum IA value of 2018 was observed for an event with no precipitation measured in the hours prior to the recording, nor during the recording itself (see Figure 8 in Section 3, Figure A4a). Nevertheless, precipitation was clearly visible in the images taken during the same time frame as the 2018 IA maximum (Figure A4b,c). Air temperature and global radiation showed a strong decrease around 2 p.m. CET and two peaks each on this day. This indicates that an intensive precipitation event occurred in the uppermost catchment, while no rain was detectable at the Jamtalhütte and at the AWS. The second IA peak later in the same day was again caused by precipitation, but this time, it was measured at the AWS.

### 4.4. Showcase Example

With robust runoff measurements available for 2020, the overall plausibility of the calculated IA can be compared to the data from the downstream runoff gauge. Figure A5 depicts the event with the maximum IA of 2020, which started with an IA peak caused by an onset of heavy precipitation. The second peak in IA is also plotted, with higher rainfall at the AWS and corresponding runoff data. This example demonstrates the capability of the method discussed here to capture the IA for extreme events of increased runoff conditions.

### 4.5. Future Improvements

The time-lapse camera setup used for this study was originally established in another research project focusing on the glacier surface [32]. However, the rapid glacier retreat caused the glacier forefield to come more and more into the camera's field of vision. With this in mind, the camera was repurposed for our specific research objectives, adding value to the existing monitoring equipment. Due to the rededication of the camera, we also had to accept some limitations in terms of image detail and the capabilities of the overall set-up. For the specific purposes of this study, it would have been beneficial to also record the lower part of the outwash plain in the image frame of the camera system. To address the issue of coverage sustainably, the complete time-lapse setup needs to be extended to a network of at least two cameras, with intersecting directions of view onto the AoI. This would not only help to reduce the effect of shadows on the analysis but also allow the results to be geo-rectified into 3D information, e.g., [33–35]. In addition, mounting the cameras with a

perpendicular axis of view would also ensure that channel structures of either direction were in the line of sight and recorded by at least one camera. For extended time series, it would also be desirable if the camera's field of view remained as constant as possible. Therefore, a more stable position of the camera systems should be provided, considering the sensitivity of maintenance work and external factors like snow pressure.

With the existing camera, we were nevertheless able to capture the extent of flooding events in the glacier forefield under most weather conditions at daytime, which could be put in a plausible context with data from a nearby AWS and a runoff gauge. We argue that this has additional benefit also for our studies on runoff, even if the previous setting was not ideal regarding the points mentioned above. Future application and use of the floodplain maps presented here may include coupling with observed topographic changes in the outwash plain. Tracking the evolution of the channel system over time and linking it to previous floodplain situation is a clear next step.

The final image analysis of this study has already been integrated into the overarching research effort, providing quantitative data by geo-rectifying and visualizing the stream properties and their evolution together with additional surveys of the outwash plain.

## 5. Conclusions

This study explored the potential of semi-automatic image analysis to detect the inundation extent and frequency from a terrestrial time-lapse camera. The pixel classification based on greyscale values from oblique hourly recordings returned plausible results of the spatial and temporal variability of surface runoff in the investigated glacier forefield. However, the presented method was strictly limited to inundation detection during daylight hours but provided reliable results for a wide range of light and weather conditions during this time of day. The weak point of the method was therefore undoubtedly (in)sufficient lighting and whether patches of snow or fog obstructed the camera's view. The image sets, processed in *ImageJ*, allowed geo-rectification to produce inundation frequency maps that provided novel insights into the evolution of the proglacial channel network over a period of three years. Meteorological and discharge data from downstream measuring stations were consulted to interpret our findings. Runoff events and their intensity were quantifiable by means frequency and spatial extension and could be attributed either to pronounced ablation, heavy precipitation or a combination of the two in the uppermost catchment. In addition, a substantial shift in the proglacial channel network over the three years was recorded, indicating persistent sediment redistribution by fluvial transport. The findings presented here and experiences with this approach encourage us to further develop and expand the camera setup for continued monitoring in data-scarce environments. Knowledge of proglacial runoff patterns will be important for understanding discharge properties and potential sediment transport in catchments with rapidly retreating glaciers and abundant sediments.

**Supplementary Materials:** The following are available online at https://www.mdpi.com/article/10.3390/w14040590/s1, Video S1: UAV footage of Jamtalferner, Scripts S1: Customized macros (ImageJ), July 2020. Table S1: Results of semi-auto analysis from 2018–2020, Table S2: Analysis of the daily maxima from 2018–2020.

**Author Contributions:** Conceptualization, C.H., L.W. and K.H.; data curation, K.H. and K.W.; methodology, C.H., L.W. and K.H.; project administration, K.H.; resources, K.H. and K.W.; software, C.H., L.W. and K.W.; supervision, S.A.; validation, C.H., K.H. and S.A.; writing—original draft, C.H. All authors have read and agreed to the published version of the manuscript.

**Funding:** This research was funded by the Austrian Academy of Sciences, Earth System Sciences (ESS) research initiative call 2018, Hidden.ice Project. The camera hardware, installation and maintenance were funded by the OeAD (Sparkling Science Project SPA05-201).

**Institutional Review Board Statement:** Not applicable.

**Informed Consent Statement:** Not applicable.

**Data Availability Statement:** The image dataset is available upon request via blackice@uibk.ac.at. Meteorological data were provided by the Hydrographic Service, Regional Government of Tyrol, accessible online under the URL: https://www.tirol.gv.at/umwelt/wasserwirtschaft/wasserkreislauf/hydrographie/, accessed on 7 February 2022. Topographic data are available from the Regional Government of Tyrol, Department of Geoinformation, accessible online under the URL: https://www.tirol.gv.at/sicherheit/geoinformation/geodaten/laserscandaten/, accessed on 7 February 2022. Discharge measurements are published Open Access on the PANGAEA repository (submitted 18 January 2022, Title: Discharge measurements in the Jamtal valley (Silvretta, Tirol, Austria) 2019, 2020, 2021). The result files of the semi-automated image analysis will be published in the Supplementary (Tables S1 and S2) to this article.

**Acknowledgments:** We extend our sincere thanks to the Regional Government of Tyrol for supplying us with meteorological and topographical data. We would also particularly like to thank Brigitte Scott for her English proofreading of our manuscript. The authors also gratefully acknowledge the publishing fund of the Vice Rectorate for Research, University of Innsbruck for financial support.

**Conflicts of Interest:** The authors declare no conflict of interest. The funders had no role in the design of the study; in the collection, analyses or interpretation of data; in the writing of the manuscript or in the decision to publish the results.

## Appendix A

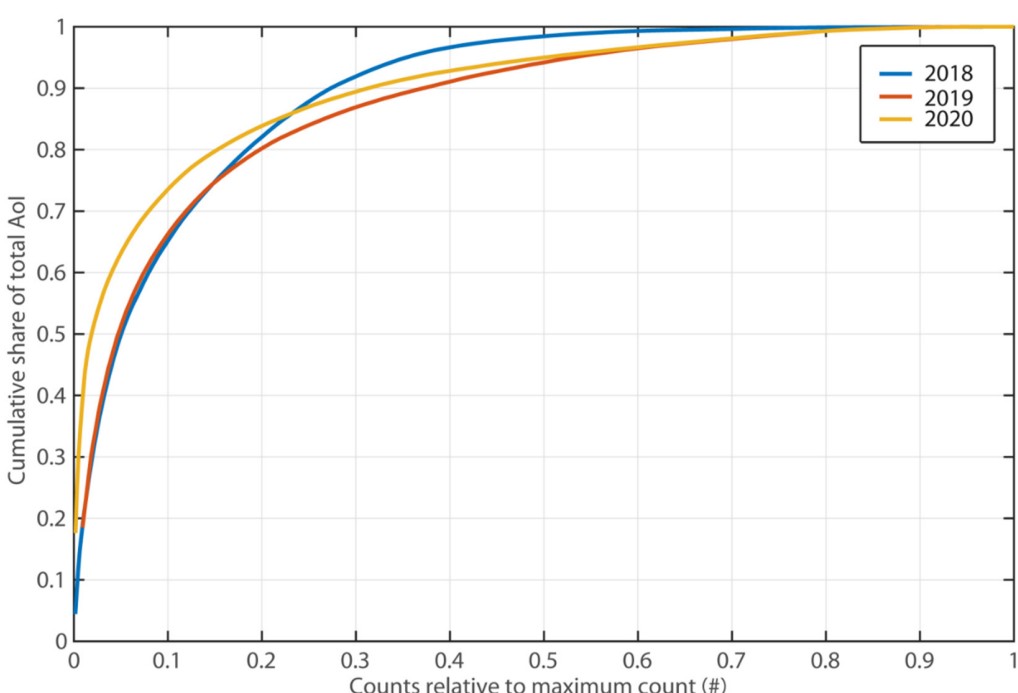

**Figure A1.** Cumulative share of the total AoI for pixels with a specific count relative to the count of the pixels most often detected as covered by water within the recording period of the respective year (maximum count).

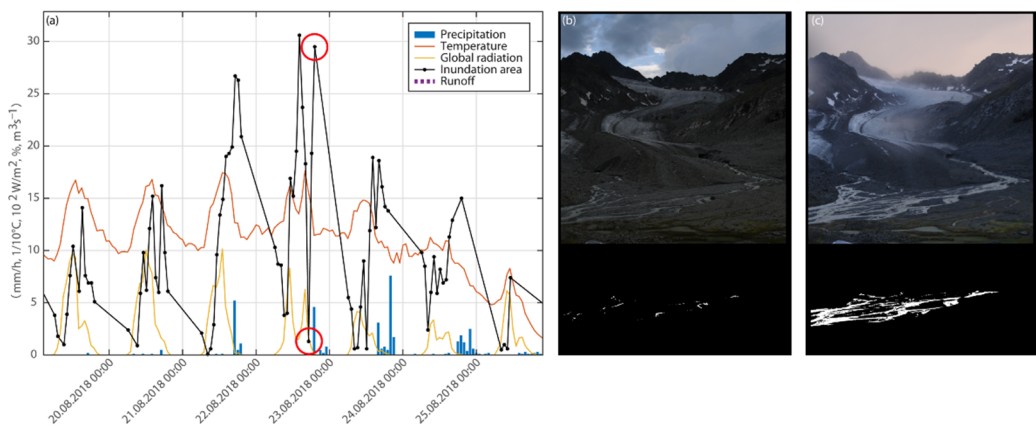

**Figure A2.** (**a**) IA and meteorological measurements for the period of interest. (**b**) shows the images in unaltered scene brightness and the detected IA for 22 August 2018 at 6 p.m. CET and (**c**) for 22 August 2018 at 8 p.m. CET, both marked by red circles in (**a**).

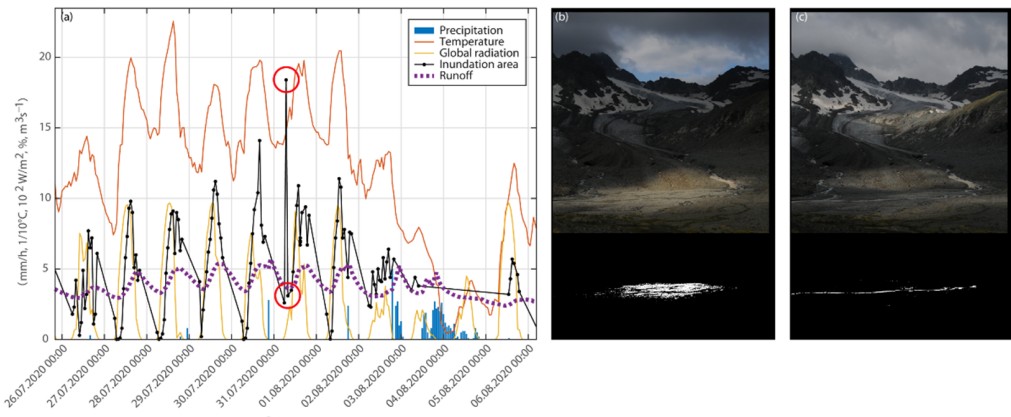

**Figure A3.** (**a**) IA and meteorological measurements for the period of interest. (**b**) shows the images in unaltered scene brightness and the detected IA for 31 July 2020 at 8 a.m. CET and (**c**) for 31 July 2020 at 9 a.m. CET, both marked by red circles in (**a**).

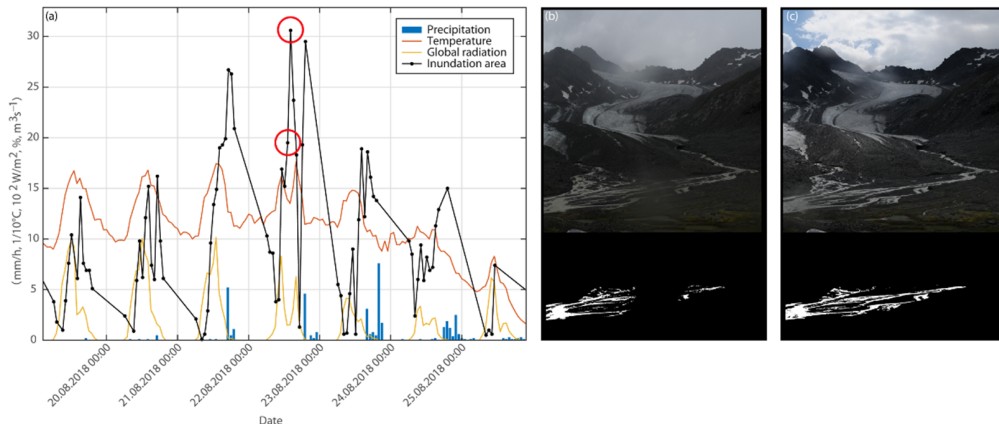

**Figure A4.** (**a**) IA and meteorological measurements for the period of interest. (**b**) shows the images in unaltered scene brightness and the detected IA for 22 August 2018 at 2 p.m. CET and (**c**) for 22 August 2018 at 3 p.m. CET, both marked by red circles in (**a**).

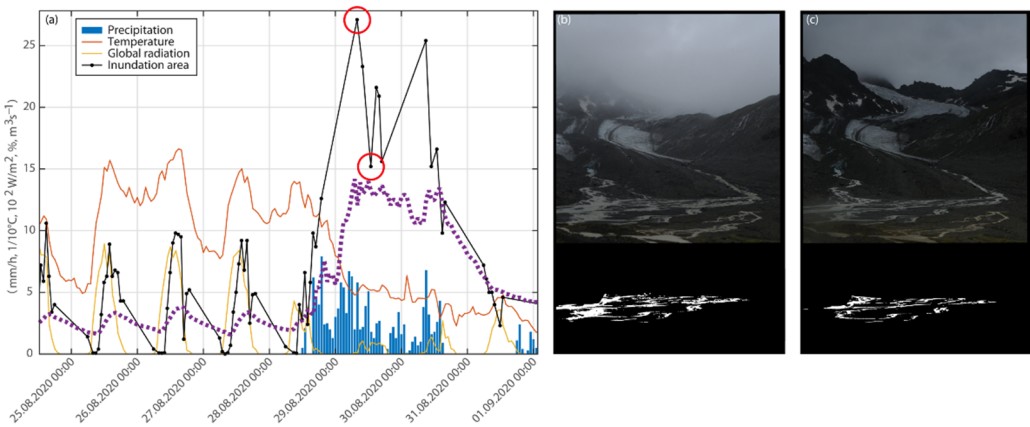

**Figure A5.** (**a**) IA and meteorological measurements for the period of interest. (**b**) shows the images in unaltered scene brightness and the detected IA for 29 August 2020 at 9 a.m. CET and (**c**) for 29 August 2020 at 1 p.m. CET, both marked by red circles in (**a**).

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
