# Peer review of "Flood Flow in a Proglacial Outwash Plain: Quantifying Spatial Extent and Frequency of Inundation from Time-Lapse Imagery"

_water, doi:10.3390/w14040590_

Round 1

Reviewer 1 Report

The research method of the manuscript is to describe the flood events that occur in mountainous areas and during the daytime with sufficient sunlight with good accuracy. However, this method is neither suitable for flood events whose duration spans daytime and night, nor for flood events that occur at night and in poor daytime conditions. Strictly speaking, the method with limited applicability used in this manuscript is not an advanced method, and is less innovative in the field of hydrology.

Some specific comments are listed as follows:

  1. Lines 70-74: The sentence is too long. Please rewrite it.
  2. Line 92: Please give out the full name for "LTER".
  3. Line 100: Are the "2141 m" and "2020 m" the elevation above sea level?
  4. Figure 1: Please give out the legend.
  5. Section 2.1: What is the capacity of the camera's compact-flash card?
  6. Line 138: It seems that the number "2010 m" should be "2020 m" based on Line 100, or vice versa.
  7. Section 2.4: Too long.
  8. Line 161: Why are "14 and 100"?
  9. Line 282: Figure 5(b) are greatly different from the other two figures. So the term "slightly" is inappropriate here, I think. The reason why Figure (b) is so different from the other two figures needs to be discussed in the Discussion section.
  10. Section of Discussion: Too short and more detail is required.

Reviewer 2 Report

Overall, the manuscript is clear, concise, and well-written. Sufficient information about previous study findings is provided. The methods that are used to analyze the data are appropriate. The presentation of the introduction, results, and discussion are satisfactory. The study and objective of the study are very relevant and important in current situations. Owing to the important topic, this paper would deserve to get published. However, I have a couple of minor suggestions that I believe should be addressed before the paper is published.

  • Images inserted in Figure A2-A5 are a little dark and hard to interpret. Please consider enhancing image quality, if possible.
  • I would recommend adding a couple of sentences about the limitations of the current study in the conclusion section.

Reviewer 3 Report

The subject is current and very important. Paraglacial transition zones in catchments with rapidly retreating glaciers and abundant sediments are key elements in high alpine river systems and promise to be revealing, yet challenging, areas of investigation for the quantification of current  and future sediment transport. The aim of this study is to capture the runoff and floodplain dynamics over a three year observation period  in this alpine proglacial outwash plain based on terrestrial timelapse imagery. The work presents the main research issues well. My basic remarks to the paper: • Literature review is correct and contains basic items. The work contains a very detailed discussion of previous studies. This is the basic advantage of this paper. • The results of the study were well analyzed. • In my opinion, the summary can be expanded a bit and refer to the results in more detail. The submitted paper made a good impression on me. It felicitously combines successes in a full-scale experiment and a calculation,  as well as a reasonable discussion of the presented results. Therefore, I have no strong remarks and I think that submitted paper can be published in its current form.
